# Ultrasonic based concrete defects identification *via* wavelet packet transform and GA-BP neural network

Tianyu Hu[1,*], Jinhui Zhao[2,*], Ruifang Zheng[1], Pengfeng Wang[3], Xiaolu Li[1] and Qichun Zhang[4]

[1] College of Mechanical and Electrical Engineering, China Jiliang University, Hangzhou, China
[2] Key Laboratory for Technology in Rural Water Management of Zhejiang province, College of Electrical Engineering,, Zhejiang University of Water Resources and Electric Power, Hangzhou, China
[3] College of Modern Science and Technology, China Jiliang Univercity, Hangzhou, China
[4] Department of Computer Science, University of Bradford, Bradford, United Kingdom
* These authors contributed equally to this work.



Corresponding author
Jinhui Zhao, jhzhao2009@zju.edu.cn

## ABSTRACT

Concrete is the main material in building. Since its poor structural integrity may cause accidents, it is significant to detect defects in concrete. However, it is a challenging topic as the unevenness of concrete would lead to the complex dynamics with uncertainties in the ultrasonic diagnosis of defects. Note that the detection results mainly depend on the direct parameters, *e.g.*, the time of travel through the concrete. The current diagnosis accuracy and intelligence level are difficult to meet the design requirement for automatic and increasingly high-performance demands. To solve the mentioned problems, our contribution of this paper can be summarized as establishing a diagnosis model based on the GA-BPNN method and ultrasonic information extracted that helps engineers identify concrete defects. Potentially, the application of this model helps to improve the working efficiency, diagnostic accuracy and automation level of ultrasonic testing instruments. In particular, we propose a simple and effective signal recognition method for small-size concrete hole defects. This method can be divided into two parts: (1) signal effective information extraction based on wavelet packet transform (WPT), where mean value, standard deviation, kurtosis coefficient, skewness coefficient and energy ratio are utilized as features to characterize the detection signals based on the analysis of the main frequency node of the signals, and (2) defect signal recognition based on GA optimized back propagation neural network (GA-BPNN), where the cross-validation method has been used for the stochastic division of the signal dataset and it leads to the BPNN recognition model with small bias. Finally, we implement this method on 150 detection signal data which are obtained by the ultrasonic testing system with 50 kHz working frequency. The experimental test block is a C30 class concrete block with 5, 7, and 9 mm penetrating holes. The information of the experimental environment, algorithmic parameters setting and signal processing procedure are described in detail. The average recognition accuracy is 91.33% for the identification of small size concrete defects according to experimental results, which verifies the feasibility and efficiency.

## INTRODUCTION

Concrete materials are widely used in modern buildings. It is a non-uniform material mixed with cement, sand, gravel and water. The random distributions of coarse aggregate and cement mortar are the causes of the heterogeneity of concrete. There exist various forms of deterioration and defects of concrete structures because of aging and environmental damage, such as internal voids (*Bien, Kaminski & Kuzawa, 2019*). Among the main health problems of concrete, surface defects are relatively easy to be detected. However, internal defects are hidden in the concrete, which is difficult to detect and is more harmful. It is significant to detect and analyze the internal defects of concrete structures in time to avoid the potential related accidents.

Commonly used methods of non-destructive testing include electromagnetic, radiological and ultrasound (*Schabowicz, 2019*). Ultrasonic has the advantages of strong penetrating power and high sensitivity, so it is mostly used in material defect detection (*Janku et al., 2019*). In actual inspection tasks, ultrasonic detection of concrete defects is based on the observation of acoustic parameters, propagation time, amplitude and main frequency of ultrasonic detection signals, etc. (*Ushakov & Davydov, 2013*; *Ozsoy, Koyunlu & Ugweje, 2017*). For example, NDT James V-C-400 V-Meter MK IV still uses the ultrasonic pulse velocity method to characterize the detection signal. These applied methods are susceptible to individual subjective factors and experience levels.

The applications of modern signal processing and artificial intelligence algorithms can achieve the automatic recognition of signals and also improve recognition efficiency and accuracy (*Tibaduiza-Burgos & Torres-Arredondo, 2015*). It is necessary to obtain effective information to characterize different types of signals before performing detection signal recognition. At present, many methods have been used to find the effective features of complex signals (*Cheema & Singh, 2019*). These signal analysis methods are mainly sparse representation, Hilbert-Huang transform, Fourier transform, wavelet transform, and so on (*Liu et al., 2021*; *Kumar & Kumar, 2019*; *Bochud et al., 2015*; *Rodriguesa et al., 2019*). However, it is difficult to extract the key information from concrete ultrasonic detection signals due to the noise influence and many mutational components, such as mutational amplitude (*Cheema & Singh, 2019*; *Combet, Gelman & LaPayne, 2012*). Among these signal preprocessing methods, wavelet transform can effectively deal with the non-stationary and high-noise complex signals. This method has been applied to process ultrasonic signals (*Acciani et al., 2010*).

Machine learning models are established with simple structures which are suitable for small sample dataset, while the scholars often choose these methods to identify detection signals (*Iyer et al., 2012*). For now, commonly used machine learning algorithms include support vector machine, neural network, etc. (*Xu & Jin, 2018*). As a class of neural networks, BP neural network (BPNN) is a classic model. It has strong nonlinear mapping ability and simple structure (*Wang, 2015*). After optimization by genetic algorithm, the

fitting ability and running speed can be improved. Note that BPNN is widely used in the field of pattern recognition, where deep learning is one of the most popular methods in pattern recognition. In the field of ultrasonic inspection, deep learning has been used to identify inspection images (*Slonski, Schabowicz & Krawczyk, 2020*). However, deep learning was rarely used to recognize concrete ultrasonic detection signals due to the high hardware performance requirements (*Shrestha & Mahmood, 2019*).

*Saechai, Kongprawechnon & Sahamitmongkol (2012)* used the support vector machine to identify the defect detection signals of concrete which obtained higher recognition accuracy. *Chen & Ma (2011)* extracted features of the weld detection signal by wavelet packet transform and used radial basis function neural network to recognize defects. *Zhang et al. (2020)* used genetic algorithm-back propagation neural network to evaluate the laser ultrasonic fault signals of uniform metal structures. The composition of the concrete selected in our paper is more complex than the research objects in the literatures. When these methods are used directly to identify concrete detection signals, the performance would be deteriorated. Therefore, a novel ultrasonic-based solution should be developed for concrete defect detection.

In this paper, we propose an intelligent method to process the ultrasonic lateral detection signals of penetrating holes in concrete. The main contributions and objective are summarized as follows:

- To improve the performance of more effective calculation and high identification accuracy, the ultrasonic detection signals are decomposed by WPT in order to extract the useful information in the detection signal. As a result, we extract the five effective features of the processed signal.
- Genetic algorithm has been used to optimize the structural parameters of the BP neural network. In the experiments with measured data, the average classification accuracy of GA-BPNN is increased by 4.66%, 4%, and 5.33% compared with BPNN, SVM, and RBF, respectively.
- This paper presents a generalized research framework on the processing and recognition of concrete ultrasonic detection signals, which lays the technical foundation for achieving the intelligent and automatic detection of concrete.

The rest of the paper has been organized as follows: In "The Process of the Proposed Algorithm", we describe the whole algorithmic procedure and principles briefly. And we present the experimental system and algorithmic parameters setting in the "Experimental Environment and Test". The test results and analysis are presented in the "Results & Discussion". We draw a conclusion in the "Conclusions and Future Work".

## THE PROCESS OF THE PROPOSED ALGORITHM

The ultrasonic pulse velocity (UPV) method is widely used in ultrasonic testing instruments which cannot meet the needs of small-size concrete defect detection. The levels of intelligence and automation of concrete testing instruments need to be improved urgently. To solve this problem, we propose a method based on WPT and GA-BPNN.

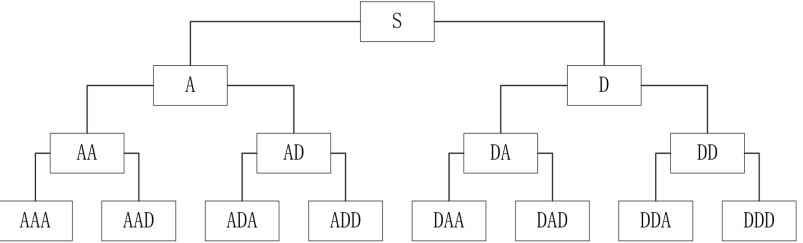

**Figure 1 Three-layer wavelet packet decomposition diagram.**

In particular, the presented algorithm in this paper consists of three parts. First, wavelet packet transform is used to attenuate noise and retain effective information from the non-stationary concrete ultrasonic detection signals. Then, the features of processed signals are extracted as the feature vector. Finally, we use the BPNN optimized by the improved GA to identify the detection signals and the K-fold cross-validation is introduced to verify the stability and generalization of GA-BPNN. We describe the main steps in the following subsections.

## Wavelet packet transform

Wavelet transform is a multi-resolution analysis method (*Babouri et al., 2019*), which is a process of using wavelet basis functions to decompose a signal into components of different frequency bands, processing the wavelet packet coefficients, and reconstructing these components into a complete signal (*Kim et al., 2020*). When using the wavelet transform to process a non-stationary signal, there are different resolutions at different locations. WPT can accurately obtain the high and low-frequency components of the signal (*Schimmack & Mercorelli, 2018*). Therefore, WPT can be considered as an effective pre-processing algorithm for feature extraction. However, the wavelet transform cannot extract the detailed information of detection signals.

### Decomposition and reconstruction of WPT

The structure diagram of the three-layer decomposition of wavelet packet is given in Fig. 1.

In Fig. 1, $S$ means an original signal. Then, $S$ can be decomposed according to the Eq. (1) to obtain $A$ and $D$. $A$ is a low-frequency component and $D$ is a high-frequency component after each decomposition of an original signal. Continuously, we decompose $A$ and $D$ in the same way. Finally, $S$ is decomposed into eight components at different frequency bands.

The basic calculation formulas of WPT are shown in Eqs. (1)–(2).

$$\begin{cases} d_j^{2n}[k] = \sum_{l \in Z} h_{l-2k} d_{j+1}^n[l] \\ d_j^{2n+1}[k] = \sum_{l \in Z} g_{l-2k} d_{j+1}^n[l] \end{cases} \tag{1}$$

$$d_j^{2n+1}[k] = \sum_{l \in Z} h_{k-2l} d_j^{2n}[l] + \sum_{l \in Z} g_{k-2l} d_j^{2n+1}[l] \tag{2}$$

where $d_{j+1}^{n}$ represents a wavelet packet coefficient sequence of the signal to be decomposed, $j$ represents a scale factor, $n$ represents the number of frequency bands, $k$ and $l$ are the positions of coefficients sequences, $d_{j}^{2n}$ and $d_{j}^{2n+1}$ represent wavelet packet coefficients sequences of signals after decomposition, $\boldsymbol{h}$ represents orthogonal real coefficients matrices of low-pass filters, and $\boldsymbol{g}$ represents orthogonal real coefficients matrices of high-pass filters.

We use the cost function to select the wavelet packet basis for the signal decomposition. At present, the Shannon entropy (*Shi et al., 2021*) is widely used where the entropy of the wavelet packet coefficient sequence $\boldsymbol{d} = \{d_j\}$ is defined by Eq. (3).

$$M(d) = -\sum_j P_j \log_2 P_j \tag{3}$$

where $P_j = \dfrac{|d_j|^2}{\|d\|^2}$, and when $P = 0$, $P\log_2 P = 0$.

### Wavelet basis function selection

How to choose the appropriate wavelet basis function is vital besides WPT decomposes ultrasonic detection signals precisely. *Samaratunga, Jha & Gopalakrishnan (2016)* think the time-frequency change of non-stationary signal is well represented by the Daubechies wavelet function in the time-frequency domain. In this paper, the db15 wavelet is selected as the base function of WPT according to the decomposition experiment analysis.

## Ultrasonic signal features selection

In pattern recognition, feature extraction is normally used for two processes: object feature data collection and classification. The quality and property of feature data greatly affect the design and the performance of pattern recognition classifiers, *e.g.*, monotonicity, which is a key problem of pattern recognition.

Scholars used wavelet coefficients after wavelet transform as feature vectors, which resulted in the very high-dimensional input data of the recognition model (*Cruz et al., 2016*). *Wan & Li (2014)* only extracted one type of feature including the energy ratio of each node after wavelet packet decomposition as the feature vector of identifying defects in carbon-fiber-reinforced polymer. *Wang et al. (2019)* selected nine features including the average peak spacing, dominant frequency, etc., to identify weld quality defects. Furthermore, scholars also choose features such as mean value, standard deviation, kurtosis, etc. as the inputs of ultrasonic detection signal recognition models (*Virmani et al., 2013*).

Based on commonly used features in the field of ultrasonic testing, we have selected useful and non-redundant features by analyzing the calculation formulas of the features and conducting experimental tests. For example, the calculation formulas and physical meaning of mean square value and energy are very similar, and they are not used as features collectively. Finally, the five features of mean value, standard deviation, kurtosis coefficient, skewness coefficient and energy ratio are retained (*Wan & Li, 2014*; *Zhang, Duffy & Orlandi, 2017*).

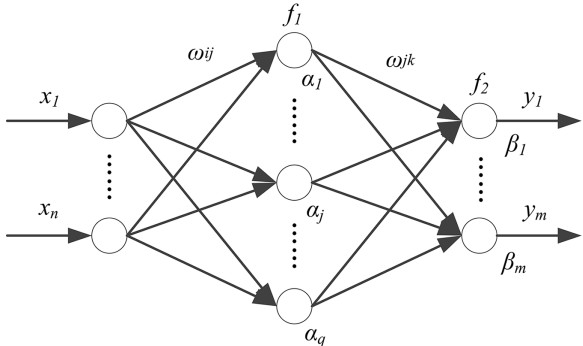

**Figure 2 BP neural network structure.**     

In order to make the feature values in the same order of magnitude and improve the convergence speed of the model, we normalize the extracted features (*Bagan et al., 2009*). The five feature values are mapped to $[-1, 1]$ and the normalized feature values are taken as input variables of the BPNN model in this paper.

## GA-BP neural network (GA-BPNN)

### BP Neural Network (BPNN)

A BPNN is made up of an input layer, a hidden layer, and an output layer. The input signal of BPNN propagates forward, and the error propagates backward. It can approximate the function of finite discontinuities (*Wang, 2015*). In addition, it has a powerful ability to deal with nonlinear problems. The structure is shown in Fig. 2.

In Fig. 2, $x$ is input data, $\omega_{ij}$ is the weight between the input layer $i$ and the hidden layer $j$, $\alpha$ is biases of the hidden layer, $f_1$ is activation functions used for the hidden layer, $\omega_{jk}$ is the weight between the hidden layer $j$ and the output layer $k$, $\beta$ is biases of the output layer, $f_2$ is the activation function used for the output layer, and $y$ is the output of the network.

### Genetic algorithm (GA)

The outputs of the BPNN are calculated according to its input-output function built on the generated weights, biases and number of hidden nodes. In this paper, the improved GA (*Peng et al., 2013*) has been used to optimize initial weights, biases and the number of hidden layer nodes. This method can make BPNN convergent fast with higher precision (*Han & Huang, 2019*).

According to the description of the improved GA (*Peng et al., 2013*), we use binary to code variables of the number of hidden layer node and use real numbers to code variables of the corresponding weights and biases for building candidate solutions in GA. We assume the maximum number of hidden layer node in the BPNN is $l$, and the number of input and output layer nodes in the network are $n$ and $m$, respectively. Then the total number of optimization variables is $1 + (l \cdot n) + l + (l \cdot m) + m$. The coding of all parameters in a candidate solution is shown in Fig. 3. For instance, if the number of hidden layer nodes is represented by a q-bit 0 to 1 string, the range of the number of hidden layer node is 0 to $2^q - 1$, and $l = 2^q - 1$.

| hidden layer nodes | $\omega_{ij}$ | $\alpha_j$ | $\omega_{jk}$ | $\beta_m$ |
|---|---|---|---|---|
| q | $n \cdot (2^q-1)$ | $2^q-1$ | $(2^q-1) \cdot m$ | m |
| binary coding | | real coding | | |

**Figure 3 Parameters coding sequence.**               

In Fig. 3, $q$ is the number of candidates hidden layer node, $l$ is the maximum number of hidden layer node, $0 < q \leq l$, $n$ is the number of nodes in the input layer, $m$ is the number of nodes in the output layer. Furthermore, errors between the actual values and the output values of the BPNN are calculated, then the reciprocal of the sum of squared errors is used as the fitness function in the GA.

In this paper, the roulette wheel method is used as the selection operator. Two individuals are selected by the selection operator. Then we use the one-point crossover method to process the binary coding arrays. The arithmetic crossover operator is used for the real number encoding sequences. For binary coding arrays, the simple mutation operator is used. We apply the non-uniform mutation operator to the real coding sequences in this paper. Briefly, we first encode the variables that need to be optimized; next, the fitness values in the initial population are calculated; then, we perform selection, crossover, and mutation operations to generate a new generation of population and obtain the maximum fitness value of each generation; finally, the optimal variable values are obtained by decoding the individual with the largest fitness value among all offspring.

## Overall steps of the WPT and GA-BPNN method

To describe the proposed concrete ultrasonic detection signal identification method, the main steps are summarized as follows.

Step1: The ultrasonic detection signal is decomposed into three layers by WPT sub-algorithm, and the wavelet packet coefficients in the main frequency node are extracted to reconstruct the signal;

Step2: The five feature variables of the reconstructed signals are calculated to establish the feature dataset;

Step3: Adopt K-fold cross-verification method to divide the dataset;

Step4: The genetic algorithm is executed to calculate the optimal configuration parameter of BPNN; select the optimal parameters of BPNN from the optimal solutions of GA, then obtain an optimized BPNN;

Step5: Use the test dataset to test the BPNN, output the recognition results.

The flowchart of the proposed method is given in Fig. 4.

## EXPERIMENTAL ENVIRONMENT AND TEST

### Experimental set-up and dataset

As an engineering application, we apply the ultrasonic transmission detection method to the practical ultrasonic detection system in which we use the P28F ultrasonic probes with the 50 kHz working frequency to generate the ultrasonic signals. An ±80-Volt square wave pulse signal is generated at the transmitting end to excite the ultrasonic probe

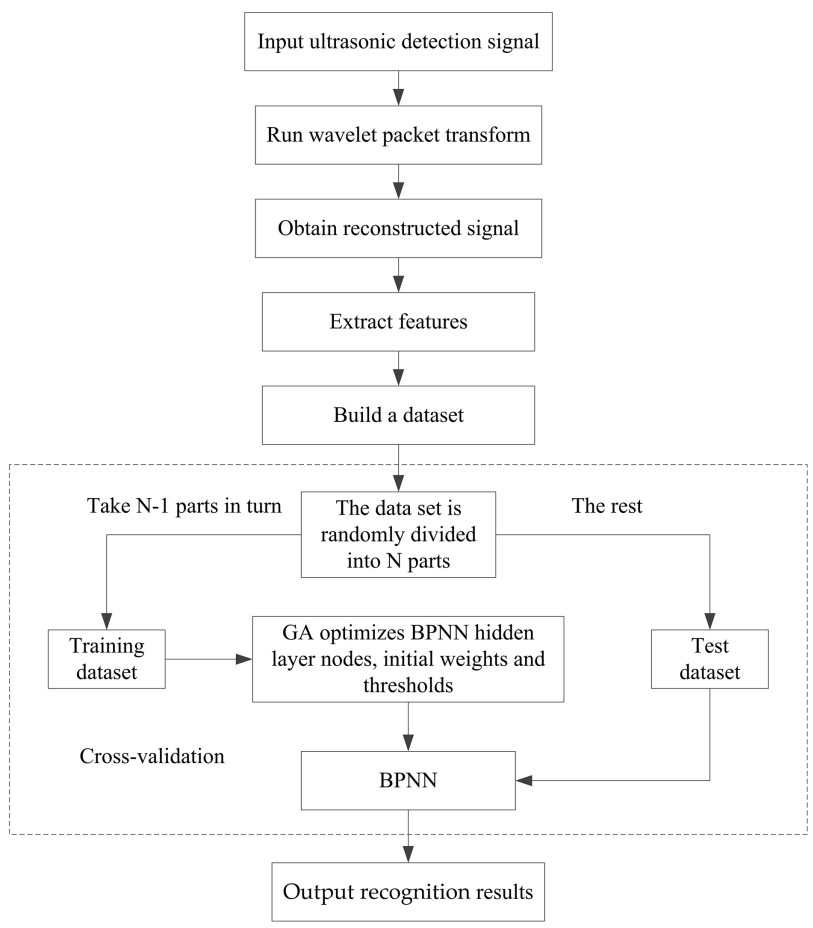

**Figure 4  Algorithm flowchat.**               

vibration. The signal sampling frequency of the receiving end is 1 MHz. The analog-to-digital conversion module used for data acquisition is 12 bits. Each detection signal used in this paper contains a total of 18,000 sampling points in six cycles. The Fig. 6 is concrete test sample. Concrete is mainly composed of cement, sand, coarse aggregate and water, the material ratio of C30 class concrete is 461, 175, 512 and 1,252 kg/m$^3$. The size of the sample block is pre-specified as follows: the length is 30 cm, the width is 20 cm and the height is 20 cm.

The experimental data are obtained by sampling repeatedly at the different positions shown in Fig. 7. The white dots of test points shown in Fig. 7 are all measured evenly. Hole defects are available in three sizes. The distance between two penetrating holes is 85 mm. The diameters of penetrating holes are 5, 7 and 9 mm, respectively. Three test points are placed on the surface of each kind of hole defect. Six test points of the defect-free structure are located between the points over the holes. The horizontal and vertical distances between the detection positions are both 45 mm. Fifteen detection positions are arranged on the concrete surface, and 10 detection data signals are obtained for each detection point. In this case study, a total of 150 ultrasonic transmission detection data samples are obtained through the experimental device in Fig. 8, including 60 sample data

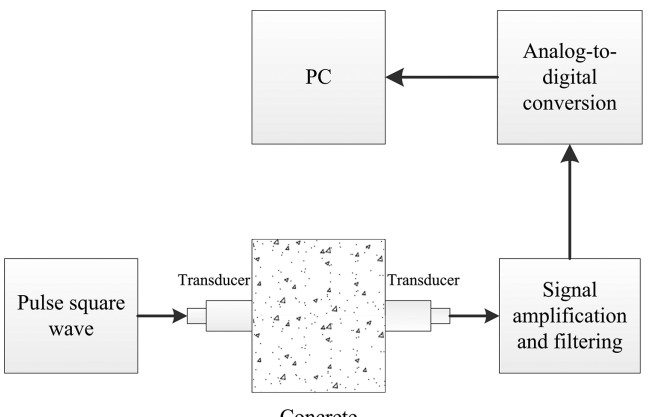

**Figure 5 The schematic of the experimental setup for evaluating concrete defects with ultrasonic.**

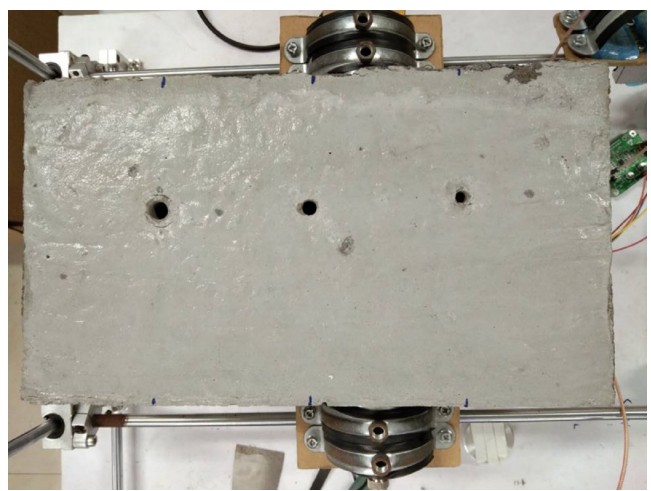

**Figure 6 Concrete test block.**

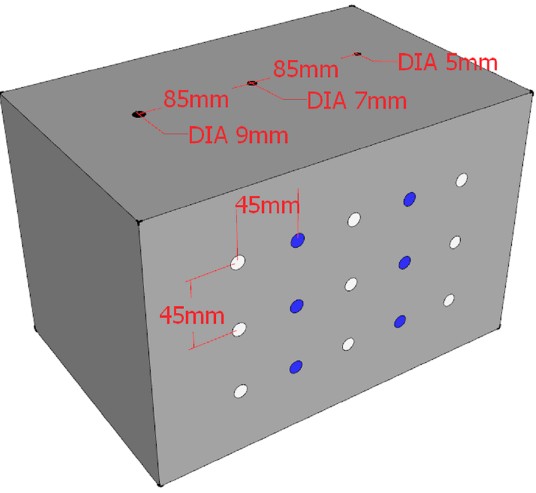

**Figure 7 Detection location diagram.**

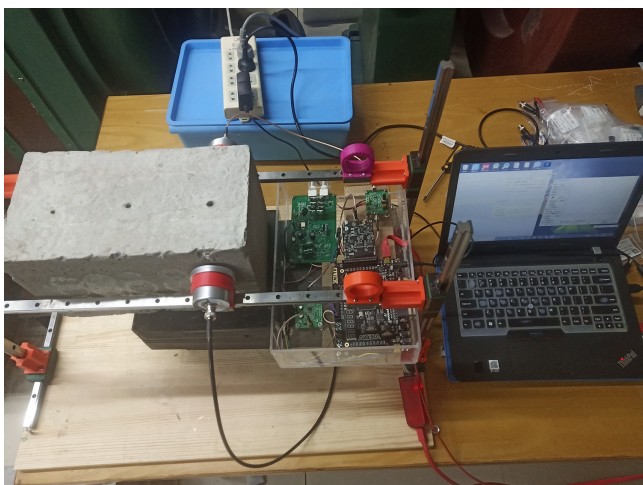

**Figure 8 Diagram of the used experimental setup.**

signals from the non-defective structure, and 90 sample data signals from the defective structure. Figure 5 shows the experimental data acquisition process of the detection system.

## Algorithm parameter settings

According to the wavelet basis function selection rule and the distribution of these detection signals' time-frequency characteristics, the db15 wavelet function is selected to perform the three-layer WPT. And most of the valid information of the signal is included in the first node of the third layer after decomposing the detection signals.

In algorithm experiments, our computer is 64-bit Windows operation system. The hardware configuration includes 2.08 GHz CPU, Inter Core i5-8400 with 6 cores, and 32 GB 2,400 MHz DDR4 memory. The application software is MATLAB R2014a version. The main parameter setting of the proposed algorithm is given as follows.

The WPT parameters setting is: the wavelet basis function is db15, the number of decomposition levels is 3, and the optimal wavelet basis Shannon entropy is selected.

The GA algorithmic parameters setting is: the maximum genetic algebra $g$ is 100, the population size $p$ is 50, the binary code length $q$ is 5, the crossover probability $Pc$ is 0.7, and the mutation probability $Pm$ is 0.05.

The BPNN algorithmic parameters setting is: the number of input nodes is 5, the number of output nodes is 2, the training stop condition is that the model error reaches 0.001 or the epochs of training reaches 1,000, and the learning rate is 0.01. Simultaneously, the cross-validation is used for training and testing the GA-BPNN model. The hidden layer function used in this paper is the hyperbolic tangent sigmoid transfer function (tansig), and the output layer function is the Log-sigmoid transfer function (logsig).

The parameters setting of the K-fold cross-validation is: K = 3, N = 150. That is, 150 samples of experimental data are randomly divided into 3 groups, and 2 groups are selected as the training data of the GA-BPNN in turn, and the remaining 1 group is used as

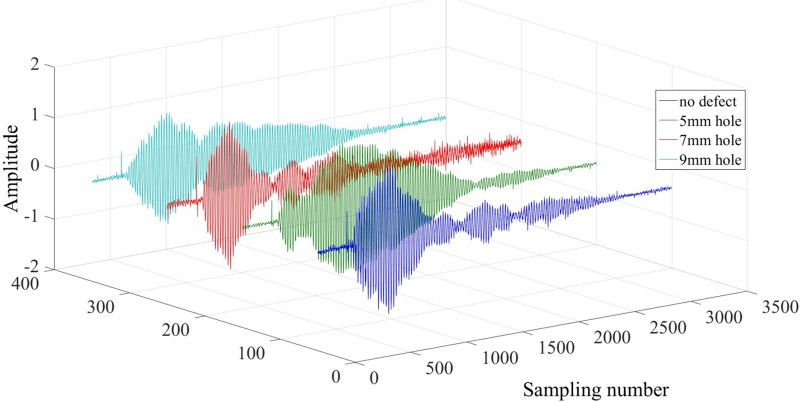

**Figure 9  Four types of ultrasonic detection signal waveforms.**

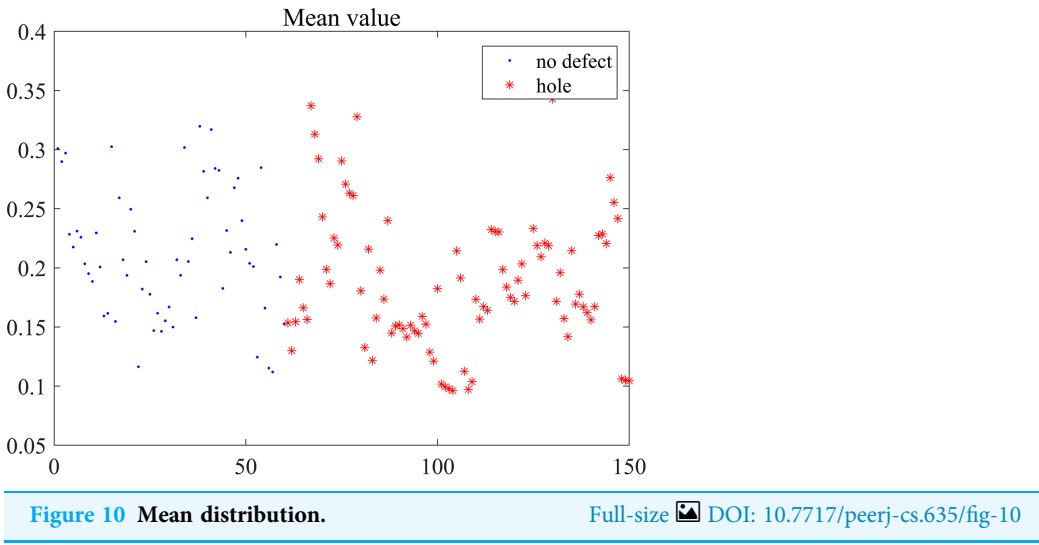

**Figure 10  Mean distribution.**

the testing data. So, the recognition rate of each test is recorded and the final result is the average of 3 recognition rates.

## RESULTS & DISCUSSION

### Experimental data analysis

Four typical waveform samples of raw detection signals are randomly selected from the experimental data, and their last period data are drawn in Fig. 9.

The figure shows the similarities and differences of the ultrasonic propagating in the concrete test block. Based on the physical mechanism of the ultrasonic propagation, the different diameters of holes are the main reason for the difference between ultrasonic detection signal waveforms. In addition, the sizes and the shapes of gravel at different locations are different in the concrete, which is another important reason for the different detection waveforms (*Garnier et al., 2009*).

Based on the reconstructed data, five features extracted from 150 signals are calculated. The five features are separately shown in Figs. 10–14.

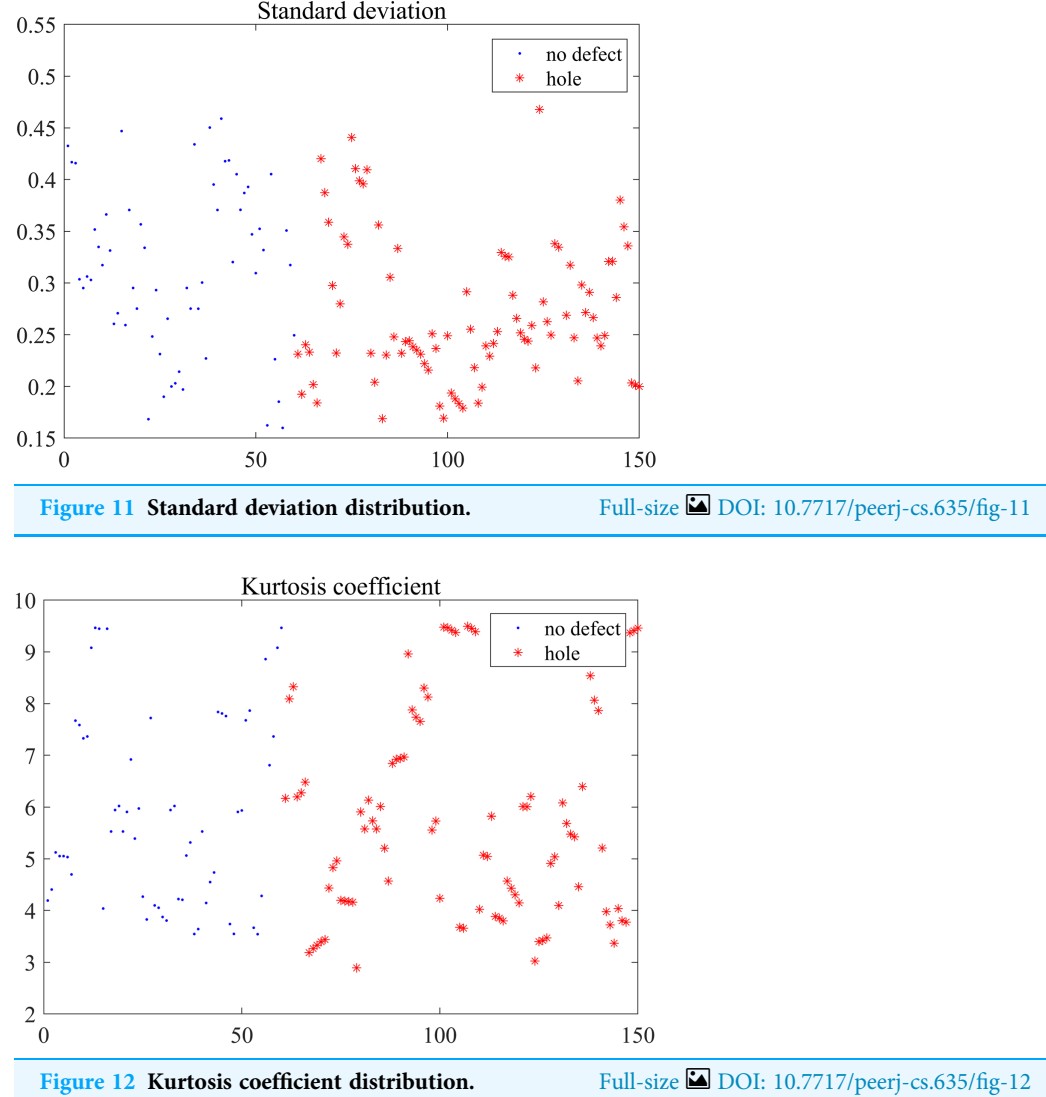

**Figure 11 Standard deviation distribution.**    

**Figure 12 Kurtosis coefficient distribution.**    

Five features of the reconstructed defective and defect-free signals do not show obvious regularity or organization from Figs. 10–14. The figures show that the feature values are different more or less even they are extracted from the same defect shared the same diameters of penetrating holes, or at the same detection points. Five features are aliasing and these reconstructed signals are inseparable linearly based on the mere measurement of single feature. On the one hand, the uneven distribution of coarse aggregate in concrete will generate acoustic measurement uncertainty, and that causes the complexity of ultrasonic detection signal. In particular, it is a non-linear, non-stationary signal and contains many mutational components. On the other hand, the stability and accuracy of the hardware system influence the output deviation, so the detection signals exist a certain distortion inevitably. Nevertheless, it can be seen that partial feature data are distributed centrally, such as the kurtosis coefficient of 9 mm defect detection data in Fig. 12. Although Different detection signals have similarities on a single feature, we can distinguish differences between different signals on multiple features fusion. Then, five

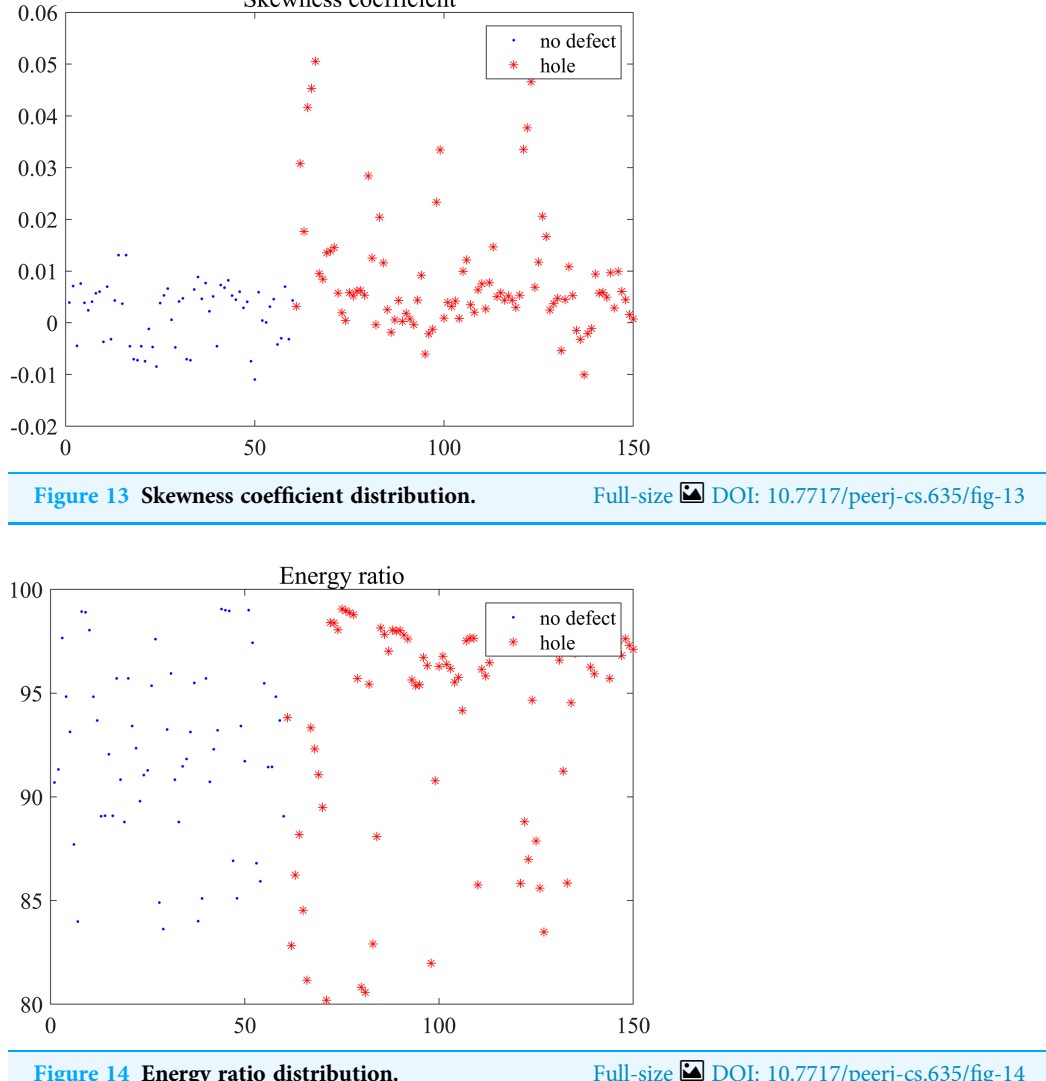

**Figure 13 Skewness coefficient distribution.**

**Figure 14 Energy ratio distribution.**

features are regarded as essential characteristics for the classification of defects in this paper.

## Comparison

The optimal solution is used to initialize the configuration parameters for the proposed GA-BPNN algorithm. The optimal number of hidden layer nodes of BPNN calculated by GA with the three-fold cross-validation method is 12, and then the number of each layer's nodes is 5, 12, and 2.

To demonstrate the advantages and disadvantages of the GA-BPNN, a BPNN without optimization is utilized for algorithmic performance analysis, and we further draw their convergent curves. We use a default function in the neural network toolbox to initialize weights and biases, with 11 hidden layer nodes, according to the empirical rule (2*5+1) in the paper of *Guo et al. (2011)*. Some parameters of the BPNN include the number of input nodes, the number of output nodes, the training stop condition, learning rate and

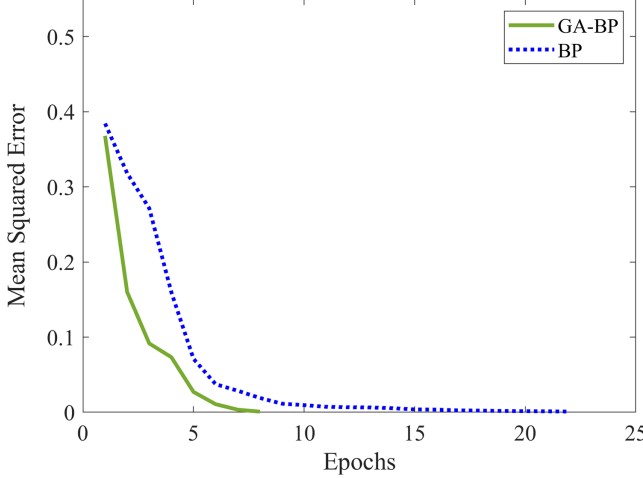

**Figure 15 The error curves in training dataset.**

activating function are the same as the parameters setting of GA-BPNN in "Experimental Environment and Test". Similarly, we use the SVM and RBF toolbox in MATLAB. The C-SVC model is chosen in the SVM algorithm, the kernel function is a polynomial, the loss parameter is 2, and the gamma parameter in the kernel function is 2.8. The target error of RBF is 0.001, the spread of radial basis functions is 1, the maximum number of neurons is 100, the number of neurons to add between displays is 1. Other parameters are default values.

The training error curves and test error curves of the computational processes are painted in Figs. 15 and 16, where two figures show how the outputs of models converge to the actual tag value. The feature data picked up for operating and drawing the curves are randomly selected from the training dataset and the test dataset respectively. The error set by the BPNN in this paper is 0.001, and the epochs required by BPNN are more than twice that of GA-BPNN. The computational cost of the BPNN is higher than that of GA-BPNN. In addition, the GA-BPNN also converges faster in the early stage of operation. In Figs. 15 and 16, it can be seen from the mean squared error curves that the GA-BPNN takes fewer epochs under the same termination conditions. The GA-BPNN has higher operating efficiency and convergence speed to approach the model's predictive values. The statistical results on 100 training data calculated by GA-BPNN with the three-fold cross-validation are shown in Table 1, the statistical results on the 50 test data are shown in Table 2. The proportion of positive and negative instances in training and test datasets are equivalent to the one in the whole dataset.

Although the convergence speed of GA-BPNN is higher, it has to spend much time to solve the optimum in the training stage, *i.e.*, it is about 489.049 s to search the optimum. Its average training time is about 0.0993 s and the average test time is about 0.0053 s. Correspondingly, the average training time of BPNN is about 0.1413 s and its average test time is about 0.0057 s. Its test recognition accuracy is about 86.67% which is less than the recognition accuracy of GA-BPNN.

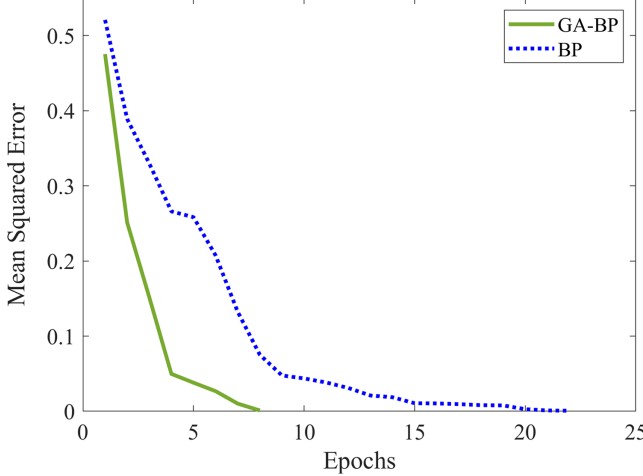

**Figure 16 The error curves in test dataset.** 

**Table 1 Training dataset recognition results.**

| No. | Class | | Recognition result | | Accuracy (%) | Average accuracy (%) |
|---|---|---|---|---|---|---|
| | | | True | False | | |
| 1 | No defect | | 38 | 2 | 96 | 95.33 |
| | Defect | 5 mm | 58 | 2 | | |
| | | 7 mm | | | | |
| | | 9 mm | | | | |
| 2 | No defect | | 36 | 4 | 95 | |
| | Defect | 5 mm | 59 | 1 | | |
| | | 7 mm | | | | |
| | | 9 mm | | | | |
| 3 | No defect | | 37 | 3 | 95 | |
| | Defect | 5 mm | 58 | 2 | | |
| | | 7 mm | | | | |
| | | 9 mm | | | | |

Three defect recognition accuracies from the three-fold cross-validation are all higher than 90% shown in the statistical results, which can prove the extracted features are effective in characterizing the presence or absence of defects in concrete, and the GA-BPNN is feasible as a concrete defect-recognition model. Furthermore, the proposed method can identify the defects automatically from detection data, then operators do not need to possess professional detection knowledge for reading and identifying recognition results. It is quite important for its practical engineering applications.

According to previous research on the recognition method of concrete ultrasonic detection signal, we choose radial basis function network (RBF) (*Chen & Ma, 2011*) and support vector machine (SVM) (*Saechai, Kongprawechnon & Sahamitmongkol, 2012*) using the data in this paper to carry out classification experiments. Also, under the 3-fold

**Table 2 Test dataset recognition results.**

| No. | Class | | Recognition result | | Accuracy (%) | Average accuracy (%) |
|---|---|---|---|---|---|---|
| | | | True | False | | |
| 1 | No defect | | 19 | 1 | 92 | 91.33 |
| | Defect | 5 mm | 27 | 3 | | |
| | | 7 mm | | | | |
| | | 9 mm | | | | |
| 2 | No defect | | 18 | 2 | 90 | |
| | Defect | 5 mm | 27 | 3 | | |
| | | 7 mm | | | | |
| | | 9 mm | | | | |
| 3 | No defect | | 20 | 0 | 92 | |
| | Defect | 5 mm | 26 | 4 | | |
| | | 7 mm | | | | |
| | | 9 mm | | | | |

**Table 3 Comparison of results of four different methods.**

| Model | Class | Sample number | Average accuracy (%) |
|---|---|---|---|
| RBF | No defect/Hole | 150 | 86 |
| SVM | No defect/Hole | 150 | 87.33 |
| BPNN | No defect/Hole | 150 | 86.67 |
| GA-BPNN | No defect/Hole | 150 | 91.33 |

cross-validation, 150 concrete ultrasonic data consisting of 5 features are used. The results of the comparative experiment are shown in Table 3. The recognition accuracy of SVM, RBF, and BPNN methods have little differences, but none of them reaches 90%. Compared with previous studies, the size of the concrete defects in this paper are smaller and therefore the detection signal is more challenging to be identified. The method we proposed is more accurate than the above three methods.

It is shown that the proposed method leads to the performance approaching high recognition accuracy. When measuring the acoustic, the degree of adhesion and contact force of the ultrasonic probe to the concrete surface may cause the recognition error due to the fact that concrete is a complex and multi-phase medium. Therefore, the obtained detection signals are complex and diverse. Although it is hard to completely identify all modes of the complex ultrasonic detection signals from concrete, more defect-type will be further investigated as our future works.

## CONCLUSIONS AND FUTURE WORK

In order to recognize the concrete defects with high reliability and accuracy by using ultrasonic testing signals, we propose an intelligent method which includes a signal processing sub-algorithm and a recognition sub-algorithm. We extract fundamental

information from the first node of the third layer by using wavelet packet transform (WPT) and calculate five feature variables of the reconstructed signals. Moreover, the GA-BPNN-based sub-algorithm identifies the concrete defects, where GA optimized BP neural network (GA-BPNN) model has been proposed embedding a K-fold cross-validation method. As a practical application of a typical type of hole defects in concrete, we utilize the method to identify the defects in a C30 class concrete test block. Based upon the test points, we obtained 150 ultrasonic detection signals containing no defect and hole defects at various locations, and then performed identification experiments based on these data sets using the method in this paper. GA-BPNN has higher diagnosis accuracy and faster running speed than existing methods. The experimental results show the effectiveness of the proposed method while the concrete hole defects have been recognized with high accuracy.

In the future, we will further verify the effectiveness of this method in more types of concrete defect (*e.g.*, cracks and foreign matter) ultrasonic detection signal data and develop more effective methods to solve complex problems in the field, such as characteristic indexes, optimizer, machine learning. Then these effective methods will be extended to more detection signal fields. Simultaneously, the sensor network solution is also our future directly for information fusion (*Naeem et al., 2021*). Simultaneously, the uneven distribution of coarse aggregate could be considered as a stochastic distribution optimization problem (*Ren, Zhang & Zhang, 2019*), its influence on the accuracy of detection signal recognition is another theoretical perspective for our future works.

## ACKNOWLEDGEMENTS

We thank two units for their help in designing the hardware system and the actual parameters testing of the ultrasonic probe, Hangzhou Ruidian Meter Co., Ltd. and Shanghai Ultra Precision Motion Control and Detection Engineering Research Center.

### Funding

This work was supported by the Zhejiang Provincial Key Research and Development Program Competitive Project under Grant (No. 2020C03074) and the Zhejiang Provincial Natural Science Foundation (ZJNSF) project under Grant (No. LY18F030012).There was no additional external funding received for this study. The funders had no role in study design, data collection and analysis, decision to publish, or preparation of the manuscript.

### Grant Disclosures

The following grant information was disclosed by the authors:
Zhejiang Provincial Key Research and Development Program Competitive Project: 2020C03074.
Zhejiang Provincial Natural Science Foundation (ZJNSF): LY18F030012.

## Competing Interests

Qichun Zhang is an Academic Editor for PeerJ.

## Author Contributions

- Tianyu Hu conceived and designed the experiments, performed the experiments, analyzed the data, performed the computation work, prepared figures and/or tables, and approved the final draft.
- Jinhui Zhao conceived and designed the experiments, performed the experiments, analyzed the data, authored or reviewed drafts of the paper, and approved the final draft.
- Ruifang Zheng conceived and designed the experiments, performed the experiments, analyzed the data, performed the computation work, prepared figures and/or tables, and approved the final draft.
- Pengfeng Wang performed the computation work, prepared figures and/or tables, and approved the final draft.
- Xiaolu Li conceived and designed the experiments, authored or reviewed drafts of the paper, and approved the final draft.
- Qichun Zhang analyzed the data, authored or reviewed drafts of the paper, and approved the final draft.

## Data Availability

The related data and codes of GA-BP, BP, SVM, and RBF are available as Supplemental Files.

## Supplemental Information

Supplemental information for this article can be found online at http://dx.doi.org/10.7717/peerj-cs.635#supplemental-information.

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
