# Peer review of "Ultrasonic based concrete defects identification *via* wavelet packet transform and GA-BP neural network"

_PeerJ Computer Science, doi:10.7717/peerj-cs.635_

## Round 0.1 · original submission · Major Revisions

Dear Dr. Zhao,

Thank you for your submission to PeerJ Computer Science.
It is my opinion as the Academic Editor for your article - Ultrasonic based concrete defects identification via wavelet packet transform and GA-BP neural network - that it requires a number of Major Revisions.

My suggested changes and reviewer comments are shown below and on your article 'Overview' screen.

Reviewer 1 ·

Basic reporting

The paper is written in a good manner. Some minor touches can improve this paper more.
The quality of the figures can be improved more
The contributions of the authors are not clear. They have mentioned in first contribution.
Several paragraphs contain trivial information and should be dropped.
I found some English mistakes please check them.
Future research directions section is core, however, it is not good at all.

Experimental design

- What are the computational resources reported in the state of the art for the same purpose?
- Please cite each equation and clearly explain its terms.
- Clearly highlight the terms used in the algorithm and explain them in the text.

Validity of the findings

- What are the evaluations used for the verification of results?

Additional comments

More recent papers must be included in all the sections and subsections.
1) Anomaly Detection in Automated Vehicles Using Multistage Attention-Based Convolutional Neural Network, IEEE Transactions on Intelligent Transportation Systems
2) Analysis of security and energy efficiency for shortest route discovery in low‐energy adaptive clustering hierarchy protocol using Levenberg‐Marquardt neural network, Transactions on Emerging Telecommunications Technologies, e3997

Reviewer 2 ·

Basic reporting

1. In the abstract, the background knowledge on the problem addressed need to be added.
2. In the abstract, the wide range of applications and its possible solutions need to be added.
3. In the abstract, the problem addressed need to be justified with more details.
4. In the Introduction section, the drawbacks of each conventional technique should be described clearly.
5. Introduction section can be extended to add the issues in the context of the existing work
6. Literature review techniques have to be strengthened by including the issues in the current system and how the author proposes to overcome the same.
7. What is the motivation of the proposed work?
8. Research gaps, objectives of the proposed work should be clearly justified.
9. The authors should consider more recent research done in the field of their study (especially in the years 2018 and 2020 onwards).
10. An error and statistical analysis of data should be performed.
11. The conclusion should state scope for future work.
12. Discuss the future plans with respect to the research state of progress and its limitations.
13. Kindly refer the below paper:
1. Rajput, D.S., Basha, S.M., Xin, Q. et al. Providing diagnosis on diabetes using cloud computing environment to the people living in rural areas of India. J Ambient Intell Human Comput (2021). https://doi.org/10.1007/s12652-021-03154-4

Experimental design

1. The authors should consider more recent research done in the field of their study (especially in the years 2018 and 2020 onwards). 6. The paper needs to provide significant experimental details to correctly assess its contribution: What is the validation procedure used?
2. Kindly provide several references to substantiate the claim made in the abstract (that is, provide references to other groups who do or have done research in this area).

Validity of the findings

An error and statistical analysis of data should be performed.

Additional comments

1. In the abstract, the background knowledge on the problem addressed need to be added.
2. In the abstract, the wide range of applications and its possible solutions need to be added.
3. In the abstract, the problem addressed need to be justified with more details.
4. In the Introduction section, the drawbacks of each conventional technique should be described clearly.
5. The introduction needs to explain the main contributions of the work more clearly.
6. The author should emphasize the difference between other methods to clarify the position of this work further.
7. The Wide ranges of applications need to be addressed in Introductions
8. The objective of the research should be clearly defined in the last paragraph of the introduction section.
9. Add the advantages of the proposed system in one quoted line for justifying the proposed approach in the Introduction section.
10. The motivation for the present research would be clearer, by providing a more direct link between the importance of choosing your own method.
11. In the introduction, the findings of the present research work should be compared with the recent work of the same field towards claiming the contribution made.
12. Introduction section can be extended to add the issues in the context of the existing work
13. Literature review techniques have to be strengthened by including the issues in the current system and how the author proposes to overcome the same.
14. The paper needs to provide significant experimental details to correctly assess its contribution: What is the validation procedure used?
15. Kindly provide several references to substantiate the claim made in the abstract (that is, provide references to other groups who do or have done research in this area).
16. An error and statistical analysis of data should be performed.
17. The conclusion should state scope for future work.
18. Discuss the future plans with respect to the research state of progress and its limitations.

---

## Round 0.2 · accepted · Accept

Dear Dr. Zhao,

Thank you for your submission to PeerJ Computer Science.

I am writing to inform you that your manuscript - Ultrasonic based concrete defects identification via wavelet packet transform and GA-BP neural network - has been Accepted for publication.

I checked the comments from Reviewer 1 and while they have asked for revisions, these are simply repeated from the first review. I found that all of their comments were addressed.

Congratulations!

Reviewer 1 ·

Basic reporting

The abstract is too long. Too much general information. Please reduce it.
The authors should define the acronym at first instance then use this in the paper. like first the authors have used this acronym (GA-BPNN) and later they defined it at the end of the abstract.
There should be starting paragraph in between these two sections " GA-BP neural network (GA-BPNN)" and "BP Neural Network (BPNN)"
- The summary at the end of the literature review should be focused on the limitations of related work.
- The authors should further add explanation about research method.

Experimental design

- What are the evaluations used for the verification of results?
- Please cite each equation and clearly explain its terms.

Validity of the findings

--

Additional comments

- The authors should add some more background and motivation in the introduction.
- Reorganize the introduction, trying to explain every word of the title.
- Authors can cite the below paper fro neural networks
Anomaly Detection in Automated Vehicles Using Multistage Attention-Based Convolutional Neural Network, IEEE Transactions on Intelligent Transportation Systems

Reviewer 2 ·

Basic reporting

The research meets all applicable standards for the ethics of experimentation and research integrity.

Experimental design

Experiments, statistics, and other analyses are performed to a high technical standard and are described in sufficient detail.

Validity of the findings

The study presents the results of original research.

Additional comments

1. The study presents the results of original research.
2. Results reported have not been published elsewhere.
3. Experiments, statistics, and other analyses are performed to a high technical standard and are described in sufficient detail.
4. Conclusions are presented in an appropriate fashion and are supported by the data.
5. The article is presented in an intelligible fashion and is written in standard English.
6. The research meets all applicable standards for the ethics of experimentation and research integrity.
7. The article adheres to appropriate reporting guidelines and community standards for data availability.